# NEURAL OPERATOR SEARCH

## ABSTRACT

Existing neural architecture search (NAS) methods explore a limited *feature-transformation-only* search space, ignoring other advanced feature operations such as feature self-calibration by attention and dynamic convolutions. This disables the NAS algorithms to discover more advanced network architectures. We address this limitation by additionally exploiting feature self-calibration operations, resulting in a heterogeneous search space. To solve the challenges of operation heterogeneity and significantly larger search space, we formulate a *neural operator search* (NOS) method. NOS presents a novel heterogeneous residual block for integrating the heterogeneous operations in a unified structure, and an attention guided search strategy for facilitating the search process over a vast space. Extensive experiments show that NOS can search novel cell architectures with highly competitive performance on the CIFAR and ImageNet benchmarks.

## 1 INTRODUCTION

Recent advances of Neural Architecture Search (NAS) are remarkable in challenging tasks, e.g. image classification (Zoph & Le, 2017), object detection (Ghiasi et al., 2019), and semantic segmentation (Liu et al., 2019a; Nekrasov et al., 2019), greatly alleviating the demands for human knowledge and interventions by automating the laborious process of designing neural network architectures. One common scheme for the standard proxy-based neural architecture search methods (Pham et al., 2018; Zoph et al., 2018; Liu et al., 2019b) is to factorise the search space via repeatedly stacking the same cell structure, within which a computing block generates an output tensor $\mathbf{F}_k$ by combining the transformations of two input feature tensors $\mathbf{F}_i$ and $\mathbf{F}_j$:

$$\mathbf{F}_k = o^{i \to k}(\mathbf{F}_i) \oplus o^{j \to k}(\mathbf{F}_j) \quad \text{s.t.} \quad i < k \ \& \ j < k, \tag{1}$$

where $o^{i \to k}$ and $o^{j \to k}$ are the $i$-th and $j$-th primitive operations for feature transformation, selected from a candidate operation set $\mathcal{O}$, and $\oplus$ is the element-wise addition. Existing NAS methods use only the standard *feature learning/transformation* operations (convolution, pooling and identity mapping) as the building components.

Besides, extensive studies (Hu et al., 2018b; Bertinetto et al., 2016; Wang et al., 2017; Jia et al., 2016; Wu et al., 2019b; Zhu et al., 2019) have proven that other advanced operations for *feature self-calibration*, such as *attention learning* and *dynamic convolutions*, can bring great benefits for representation learning. For example, Hu et al. (2018b) proposes Squeeze-and-Excitation Networks to explicitly model inter-dependencies between channels by learning channel-wise self-attention. Jia et al. (2016) presents Dynamic Filter Networks to generate context-aware filters for increasing the flexibility and adaptiveness of networks. However, these useful feature calibration elements have *never* been well exploited in NAS, significantly limiting the potentials of NAS which aims for automatically discovering more sophisticated and advanced network architectures without human engineering.

In this work, we aim to address this limitation by extending the search space of NAS with feature self-calibration operations for scaling up the search boundary. This makes a *heterogeneous search space*. Consequently, the way of feature tensor interaction and combination is dramatically diversified, from the conventional addition operator $\oplus$ only to the combination of addition $\oplus$, multiplication $\odot$ for attention modelling, and dynamic convolution $\circledast$. In this regard, we call the proposed method **Neural Operator Search** (NOS).

Such a search space enhancement is critical since NAS is enabled to explore stronger and previously undiscovered network architectures, which opens a door to potentially take the NAS research to the

next level. In the *no free lunch* saying, this also comes with two new challenges: (i) It is non-trivial and more challenging to assemble such heterogeneous tensors and operations (i.e. features, attentions and dynamic weights) in a unified computing block, as compared to the conventional homogeneous feature-tensor-to-feature-tensor transformation; (ii) The search space increases exponentially which leads to a much harder NAS problem.

To address the first challenge, we formulate a *heterogeneous operator cell* characterised by a novel heterogeneous residual block. This block, formulated in a residual learning spirit (He et al., 2016), is designed specially for fusing all the different types of tensors and operations synergistically. To solve the second challenge, we propose leveraging the *attention transfer* (Zagoruyko & Komodakis, 2017) idea to facilitate the search behaviour across this significantly larger network space via following the attention guidance of a pretrained teacher model. As we will show, this guidance not only makes the search more efficient but also improves the search result.

Our **contributions** in this work are: **(1)** We present a novel heterogeneous search space for NAS characterised by richer primitive operations including both conventional feature transformations and newly introduced feature self-calibration. This breaks the conventional selection limit of candidate neural networks and enables the NAS process to find stronger architectures, many of which are impossible to be discovered in the conventional space. This opens new territories for supporting stronger NAS algorithms and new possibilities for most expressive architectures ever to be revealed. **(2)** We formulate a novel Neural Operator Search (NOS) method dedicated for NAS in the proposed heterogeneous search space, with a couple of key designs – heterogeneous residual block for fusing different types of tensor operations synergistically and attention guided search for facilitating the search process over a vast search space more efficiently and more effectively. **(3)** With extensive comparisons to the state-of-the-art NAS methods, the experiments show that our approach is highly competitive on both CIFAR and ImageNet-mobile image classification tests.

## 2 RELATED WORK

**Neural Architecture Search.** Since the seminal work by Zoph & Le (2017), neural architecture search has gained a surge of interest, effectively replacing laborious human designs by the computational process. From the *strategy* point of view, NAS methods can be categorised into two types: (1) proxy-based (Zoph & Le, 2017; Zoph et al., 2018; Pham et al., 2018; Liu et al., 2019b) and (2) proxy-less (Cai et al., 2019; Tan et al., 2019; Wu et al., 2019a) NAS. Specifically, to alleviate the computational cost during search, the proxy-based NAS methods search for building cells on proxy tasks, with one or more of following compromised strategies: starting with fewer cells; using a smaller dataset (e.g. CIFAR-10); learning with fewer epochs. Then, to transfer to the large-scale target task, one can build a network by stacking searched cells without further exploration. However, suffering from lacking of directness and specialisation, the searched cells by proxy-based NAS methods are not guaranteed to be optimal on the target task. In contrast, proxy-less NAS methods directly learns architectures on a target task by starting with an over-parameterised network (*supernet*) that contains all possible paths, in which the redundant paths are pruned to derive the optimised architecture. Notwithstanding significant better results than proxy-based approaches, proxy-less NAS methods require massive computational cost and GPU memory assumption, due to learning with the vast-size *supernet*. From the *optimisation* point of view, existing NAS methods usually fall into three groups: reinforcement learning (RL) based methods, evolutionary algorithm (EA) based methods, and gradient differentiable (GD) methods. In particular, RL-based NAS methods (Zoph & Le, 2017; Pham et al., 2018; Tan et al., 2019) control the selection of architecture component in a sequential order with policy networks. EA-based NAS methods (Real et al., 2019; Liu et al., 2018b) employ the validation accuracies to guide the evolution of a population of initialised architectures. RL- an EA-based NAS methods usually suffer from low efficiency and high computational resource demand, due to the fundamental searching challenge in a discrete space. In contrast, GD-based NAS methods (Liu et al., 2019b; Xie et al., 2019; Luo et al., 2018) conduct searching over a continuous space by relaxation or mapping, substantially reducing the search cost to a few GPU days. Whilst varying in the algorithmic aspects, all these works commonly explore the *feature-transformation-only* search spaces without more diverse and advanced operations as we investigate here. To show the NAS potential of the proposed richer search space with self-calibration learning operations, we take the efficient proxy-based GD optimisation due to the resource constraint.

**Self-Calibration.** Self-calibration is a type of mechanism enabling a network to dynamically perform input-conditional self-adjustment, which has been studied extensively in both the computer vision (Hu et al., 2018b; Jia et al., 2016; Li et al., 2018; Park et al., 2018) and natural language processing (NLP) literature (Wu et al., 2019b; Vaswani et al., 2017). There are two typical paradigms of self-calibration: *self-attention learning* and *dynamic convolutions*, realised via an *element-wise multiplication* operator $\odot$ and a *dynamic convolution* operator $\circledast$, respectively. Despite showing significant efficacy, self-calibration is only exploited *independently* after architecture hand-design (Hu et al., 2018b) or auto-search (Tan et al., 2019). We move a step further by fully exploring the potential of self-calibration along with feature transformation in joint optimisation, bringing a richer search space for neural architecture search.

**Knowledge Distillation.** There are recent works that use knowledge distillation to help computer vision and NLP tasks. Three types of knowledge are usually considered in distillation: features (Yim et al., 2017), attention (Zagoruyko & Komodakis, 2017), and predictions (Hinton et al., 2015). We leverage the attention distillation with a different objective – alleviating the intrinsic training-test discrepancy issue of the proxy-based NAS strategy, particularly with a more expressive search space. This represents a novel exploitation of attention distillation (Zagoruyko & Komodakis, 2017).

## 3 METHOD

In this section, we start by formulating a *heterogeneous search space* for NAS (Section 3.1), followed by a dedicated *heterogeneous operator cell* to enable composing the heterogeneous operations in a unified computing block with synergistic interaction and cooperation (Section 3.2). To overcome the intrinsic architecture discovery challenges from more expressive search space, we further develop an *attention guided search* scheme (Section 3.3).

### 3.1 HETEROGENEOUS SEARCH SPACE

To enrich the NAS search space so that more advanced network architectures can be discovered, we introduce a *heterogeneous search space* $\mathbb{A}$ that considers three different types of representation learning capabilities: (1) *Feature transformations*; (2) *Attention learning*; and (3) *Dynamic convolutions*. More concretely, we form three sets of primitive computing operations that produce *features*, *attentions* and *dynamic weights*, respectively. This novel search space generalises the conventional counterpart which is limited to the first type of operations (Liu et al., 2019b; Pham et al., 2018), and incorporates the self-calibration learning capabilities (i.e. the second and third types) in NAS. Importantly, while the search space changes, the generic search strategies still apply therefore being largely open for collaborating with existing NAS methods. For instance, in the proxy-based NAS strategy we may first search for a computing cell with heterogeneous operations as the building block and then form the final network architecture by sequentially stacking multiple such cells layer-by-layer.

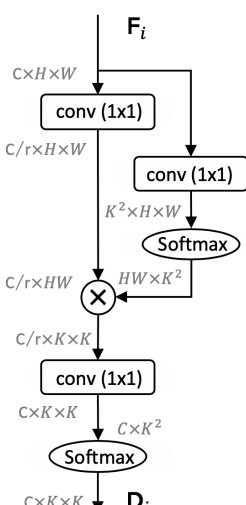

Figure 1: Structure of the proposed dynamic convolutions for image classification. $\otimes$ denotes matrix multiplication.

Next, let us describe the heterogeneous primitive operation set $\mathcal{O}$ which consists of the following three disjoint subsets: $\mathcal{O}_f$, $\mathcal{O}_a$ and $\mathcal{O}_d$, along with their aggregation or application operators.

**Feature Transformation Operations $\mathcal{O}_f$.** We adopt the feature transformation/learning operation set $\mathcal{O}_f$ same as in Liu et al. (2019b; 2018a), including the following 7 operations: $3 \times 3$ and $5 \times 5$ separable convolutions, $3 \times 3$ and $5 \times 5$ dilated separable convolutions, $3 \times 3$ average pooling, $3 \times 3$ max pooling, and identity. Every operation $o_f \in \mathcal{O}_f$ takes as input a feature tensor and

outputs another feature tensor, i.e. *homogeneous* feature-tensor-to-feature-tensor transformation. For multiple feature tensor aggregation, the element-wise addition operator $\oplus$ is typically used.

**Attention Learning Operations** $\mathcal{O}_a$. Inspired by recent exquisite designs of attention learning modules (Hu et al., 2018b; Li et al., 2018; Park et al., 2018), we form the $\mathcal{O}_a$ by considering two types of attention learning prototypes: *spatial-wise* and *channel-wise* attentions. Specifically, a *spatial-wise* attention operation learns a saliency map for an input feature tensor in order to calibrate the importance of different spatial positions. In contrast, a *channel-wise* attention operation produces a vector of scaling factors from the aggregated global context of an input tensor for adaptively calibrating the channel dependency. To enforce attentive calibration on feature tensor, the element-wise multiplication operator $\odot$ is a typical choice for both *spatial-wise* and *channel-wise* attentions.

**Dynamic Convolution Operations** $\mathcal{O}_d$. Dynamic convolutions, designed for the sake of self-adaptation, *generate* dynamic kernel weights in accordance with the input feature tensor. It is often in form of depth-wise separable convolution as the feature transformation operation. Tailored for either NLP or dense prediction tasks, existing dynamic convolution designs (Wu et al., 2019b; Jia et al., 2016) are not suitable for image classification (our focus) with different problem nature. It hence needs to be reformulated in order to be effective for learning discriminative image representations. We consider two design principles: (i) structurally lightweight whilst (ii) functionally strong with great modelling capability.

To that end, we present an exquisite dynamic convolution structure specialised for cost-effective image classification, as shown in Fig. 1. Concretely, it consists of three compact modules composed in an exquisite cooperation: (a) a *bottleneck* module, to compress an input feature tensor by a ratio of $r$; (b) a *kernel transform* module, to learn latent representations with a kernel dimension of $k \times k$; (c) a *kernel decode* module, to read out the dynamic kernel weights with the channel dimension same as the input feature tensor. This design is motivated, in part, by the long-range dependency modeling (Wang et al., 2018; Cao et al., 2019) and global context aggregation (Hu et al., 2018b;a), elegantly integrating their merits via a unified formulation. For the output of dynamic convolutions, we consider two common kernel sizes: $3 \times 3$ and $5 \times 5$. In a depth-wise manner, we apply a standard or dilated convolution operator $\circledast$ to transform the input feature tensor. It is noteworthy to point out that, this type of convolutional kernel is *specific* for each feature tensor of a particular image sample (i.e. dynamic), rather than learned from a training dataset and *fixed* for all the input samples (i.e. static) as the conventional convolutional operations in the feature transformation set.

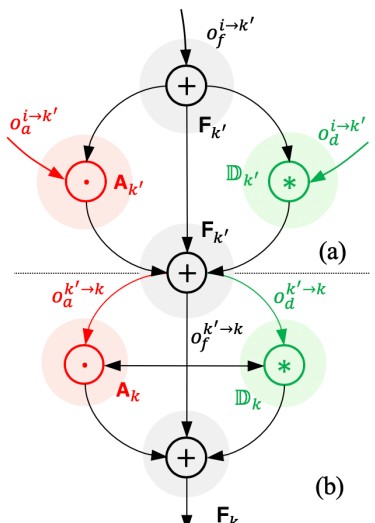

Figure 2: Heterogeneous Residual block for formulating the inner node computation. (a) First-tier *individual* computation; (b) Second-tier *collective* computation.

Detailed implementations of self-calibration operations are presented in Appendix A.1.

## 3.2 HETEROGENEOUS OPERATOR CELL

Due to different natures of heterogeneous computing capabilities, a *unification* structure is needed for composing the primitive operations $\mathcal{O} = \mathcal{O}_f \cup \mathcal{O}_a \cup \mathcal{O}_d$ and aggregation/application operators $\mathcal{C} = \{\oplus, \odot, \circledast\}$ in such a way that their representation learning potentials can be well mined. To that end, we formulate a *heterogeneous operator cell*, a directed acyclic graph (DAG) $\mathcal{G} = (\mathcal{V}, \mathcal{E})$, joining conventional feature transformations and proposed self-calibration operations synergistically.

Formally, a heterogeneous operator cell consists of $N$ ordered feature (tensor) nodes $\mathcal{V} = \{\mathbf{F}_k |, 1 <= k <= N\}$. Following (Zoph et al., 2018), $\mathbf{F}_1$ and $\mathbf{F}_2$ are the outputs from the previous cells regarded as two *input nodes*, $\{\mathbf{F}_k\}_{k=3}^{N-1}$ denotes the *inner nodes* that perform computation,

and the $N$-th node $\mathbf{F}_N$ is the cell *output node* formed as the concatenation of all the inner nodes, i.e. $\mathbf{F}_N = \mathtt{concat}(\{\mathbf{F}_k\}_{k=3}^{N-1})$. The *edge* $e^{i \to k} = (i, k) \in \mathcal{E}$ specifies the connection between the $i$-th and $k$-th nodes (the information flow $i \to k$), associated with a specific operation $o^{i \to k}$ selected from the heterogeneous primitive operation set $\mathcal{O}$. The key is to design a computing block for the inner nodes with heterogeneous computations.

**Heterogeneous Residual Block.** It is non-trivial to design a heterogeneous computing block due to being *not* straightforward feature-tensor-to-feature-tensor transformation as in the conventional homogeneous operation. It involves self-calibrating the input feature tensor *itself* in addition to the homogeneous feature transformation. To facilitate adding the extra capacity, we formulate a heterogeneous residual block (see Fig. 2) characterised by a *surrogate node* $k'$ in the computing block associated with each inner node $k$, for enabling richer feature tensor manipulations. This is in a residual learning spirit (He et al., 2016), allowing to conduct self-calibration reliably.

Moreover, we design a two-tier computing hierarchy: the first tier for *individual* computation per input feature tensor to capture the specificity, and the second tier for *collective* computation on the set of all the input feature tensors as a whole to capture the intrinsic structural relations between feature tensors and the global input properties. The two tiers are connected by the surrogate node $k'$.

Formally, we take as input all the previous nodes $\{\mathbf{F}_i |, i < k\}$, process them separately with heterogeneous operations, and combine the processed results by summation (Fig. 2 (a)):

$$\mathbf{F}_{k'} = \sum_{i<k} o_f^{i \to k'}(\mathbf{F}_i), \quad \mathbf{A}_{k'} = \sum_{i<k} o_a^{i \to k'}(\mathbf{F}_i), \quad \mathbb{D}_{k'} = \left\{ o_d^{i \to k'}(\mathbf{F}_i) \right\}_{i<k} \tag{2}$$

where $\mathbf{F}_{k'}$, $\mathbf{A}_{k'}$, and $\mathbb{D}_{k'}$ are the three types of intermediate outputted tensors, i.e. features, attentions, and dynamic weights, respectively. These are subsequently aggregated into an *intermediate calibrated tensor*, i.e. the surrogate node $\mathbf{F}_{k'}$, using element-wise addition in-between on feature self-calibration and transformation as:

$$\mathbf{F}_{k'} = \underbrace{\mathbf{F}_{k'}}_{feature} \oplus \underbrace{(\mathbf{F}_{k'} \odot \mathbf{A}_{k'})}_{attention} \oplus \underbrace{\sum_{\mathbf{D}_{k'} \in \mathbb{D}_{k'}} \mathbf{F}_{k'} \circledast \mathbf{D}_{k'}}_{dynamic\ conv} \tag{3}$$

Next, $\mathbf{F}_{k'}$ is used as the input for the second-tier set-level collective computation (Fig. 2 (b)). Likewise, we consider the same three types of operations:

$$\mathbf{F}_k = o_f^{k' \to k}(\mathbf{F}_{k'}), \quad \mathbf{A}_k = o_a^{k' \to k}(\mathbf{F}_{k'}), \quad \mathbb{D}_k = \left\{ o_d^{k' \to k}(\mathbf{F}_{k'}) \right\}, \tag{4}$$

and form the inner node $\mathbf{F}_k$ via further feature self-calibration and transformation as:

$$\mathbf{F}_k = \underbrace{\mathbf{F}_k}_{feature} \oplus \underbrace{(\mathbf{F}_k \odot \mathbf{A}_k)}_{attention} \oplus \underbrace{\sum_{\mathbf{D}_k \in \mathbb{D}_k} \mathbf{F}_k \circledast \mathbf{D}_k}_{dynamic\ conv} \tag{5}$$

In doing so, our heterogeneous residual block presents a two-tier combinatorial operations structure for each inner node, resulting in a more expressive search space (see Section 4.2).

### 3.3 ATTENTION GUIDED SEARCH OPTIMISATION IN A HETEROGENEOUS SEARCH SPACE

To showcase the effectiveness of the proposed heterogeneous search space and operator cell, we adopt the proxy-based NAS strategy, due to the computing resource constraints and the enormous search space. This search is done by constructing a small proxy network parametrised by $\Theta$.

**Attention Guided Search.** Compared with proxyless search strategy, proxy-based NAS is more efficient but relatively less optimal due to *not* directly optimising the final network architecture. This training-test discrepancy problem can be worsened when the search space provides more flexibility and combinatorial capability, such as the proposed space. To solve this obstacle, we propose attention guided search, which optimises the proxy network in a knowledge distillation manner by injecting an external guidance from a pre-trained teacher network into the NAS process.

Specifically, we leverage the attention transfer idea (Zagoruyko & Komodakis, 2017) that encourages a student (the proxy network in our case) to hierarchically imitate a teacher's hidden attention knowledge. Intuitively, this may benefit the search for self-calibration learning. Formally, let us denote a feature tensor at the $j$-th stage of the teacher and student network as $\mathbf{F}_T^j$ and $\mathbf{F}_S^j$, separately. Attention transfer is realised by imposing an alignment loss function across the two networks as:

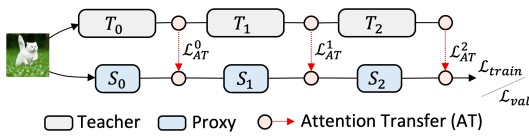

Figure 3: Overview of attention guided search. $T_i$ and $S_i$ ($i \in \{0, 1, 2\}$) denote the $i$-th stage of the teacher and proxy (student) networks.

$$\mathcal{L}_{AT} = \frac{1}{2} \sum_{j \in \mathcal{J}} \| \frac{\boldsymbol{x}_S^j}{\|\boldsymbol{x}_S^j\|_2} - \frac{\boldsymbol{x}_T^j}{\|\boldsymbol{x}_T^j\|_2} \|_2, \text{ with } \boldsymbol{x}_{S/T}^j = vec(\sum_i |\mathbf{F}_{S/T}^j(\cdot, \cdot, i)|^2) \quad (6)$$

where $\boldsymbol{x}_{S/T}^j$ is the spatial-wise accumulated feature vector. An overview of attention guided search is depicted in Fig. 3.

**Optimisation.** For NAS optimisation, we adopt the DARTS method (Liu et al., 2019b). In our context, we conduct the continuous relaxation over all the possible heterogeneous operations $\mathcal{O}$ for making a continuous search space:

$$\bar{o}^{i \rightarrow j}(x) = \sum_{o \in \mathcal{O}} \frac{\exp\left(\boldsymbol{a}_o^{i \rightarrow j}\right)}{\sum_{o' \in \mathcal{O}} \exp\left(\boldsymbol{a}_{o'}^{i \rightarrow j}\right)} o(x), \quad (7)$$

where an architecture vector $\boldsymbol{a}_o^{i \rightarrow j} \in \mathbb{R}^{|\mathcal{O}|}$ is used for each possible connection $i \rightarrow j$. We summarise the architecture vector of all the connections as a matrix $\boldsymbol{A} = \left[\boldsymbol{a}^1, \cdots, \boldsymbol{a}^{|\mathcal{E}|}\right] \in \mathbb{R}^{|\mathcal{E}| \times |\mathcal{O}|}$. With this relaxation, we can jointly optimise the architecture parameters $\boldsymbol{A}$ and the network weights $\boldsymbol{\Theta}$ in a fully gradient differentiable manner. Equipped with the proposed attention guidance search, the search objective function is finally formulated as the following bilevel optimisation process:

$$\boldsymbol{\Theta}^*(\boldsymbol{A}) = \arg \min_{\boldsymbol{\Theta}} \mathcal{L}_{train}(\boldsymbol{\Theta}, \boldsymbol{A}) + \lambda \mathcal{L}_{AT}(\boldsymbol{\Theta}, \boldsymbol{A}), \quad (8)$$

$$\boldsymbol{A}^* = \arg \min_{\boldsymbol{A}} \mathcal{L}_{val}(\boldsymbol{\Theta}^*(\boldsymbol{A}), \boldsymbol{A}) + \lambda \mathcal{L}_{AT}(\boldsymbol{\Theta}^*(\boldsymbol{A}), \boldsymbol{A}), \quad (9)$$

where $\lambda$ denotes the weighting hyper-parameter. For the first level Eq. (8), we learn the optimal parameters $\boldsymbol{\Theta}^*$ for a given architecture $\boldsymbol{A}$ w.r.t a training objective $\mathcal{L}_{train}$ and the attention alignment loss $\mathcal{L}_{AT}$. The second level Eq. (9) then explores the optimal architecture $\boldsymbol{A}^*$ over the heterogeneous search space $\mathbb{A}$ w.r.t a validation objective $\mathcal{L}_{val}$ and $\mathcal{L}_{AT}$. For image classification, $\mathcal{L}_{train}$ and $\mathcal{L}_{val}$ usually take the cross-entropy loss function.

**Search Outcome.** Once the above alternated optimisation is done, we derive an amenable cell architecture with heterogeneous operators. In practice, for each heterogeneous computing block we retain the top-2 strongest incoming operations with at least one feature transformation operation for the first-tier (Fig. 2(a)), and the top-1 strongest operation for the second-tier (Fig. 2(b)).

## 4 EXPERIMENTS

We evaluated the proposed NOS method on image classification using three common datasets. **CIFAR10/100:** Both CIFAR10 and CIFAR100 have 50K/10K train/test RGB images of size $32 \times 32 \times 3$, categorised into 10 and 100 classes, respectively (Krizhevsky et al., 2009). **ImageNet:** We use the ILSVRC2012 version for large-scale image classification evaluation, containing 1.28M training images, 50K validation samples, and 1K classes (Russakovsky et al., 2015).

We first conduct preliminary experiments on CIFAR10/100 to select the heterogeneous primitive operations $\mathcal{O}$. To test the efficacy and transferability of NOS, we search the cell structures on CIFAR10 only, and compare the performance with existing methods on CIFAR10/100 and ImageNet.

Table 1: Evaluating the feature self-calibration operations on CIFAR10 and CIFAR100.

| Model | Type | Kernels | CIFAR10 | | CIFAR100 | | FLOPS(M) | #Params(MB) |
|-------|------|---------|---------|---------|----------|----------|----------|-------------|
| | | | Top-1(%) | Top-5(%) | Top-1(%) | Top-5(%) | | |
| ResNet-18 | - | - | 4.95 | 0.22 | 23.61 | 7.16 | 555.42 | 11.17 |
| + Dynamic | Normal | 3 | 4.63 ↑ | 0.13 ↑ | 22.63 ↑ | 6.44 ↑ | + 3.85 | + 0.03 |
| | | 5 | 4.78 ↑ | 0.14 ↑ | 23.45 ↑ | 6.82 ↑ | + 7.62 | + 0.04 |
| | Dilated | 3 | 4.97 ↓ | 0.23 ↓ | 24.00 ↓ | 7.28 ↓ | + 3.85 | + 0.03 |
| | | 5 | 4.92 ↑ | 0.17 ↑ | 23.75 ↓ | 7.20 ↓ | + 7.62 | + 0.04 |
| + Attention | Spatial | | 4.79 ↑ | 0.16 ↑ | 23.51 ↑ | 7.04 ↑ | + 1.08 | + 0.01 |
| | Channel | | 4.83 ↑ | 0.19 ↑ | 23.20 ↑ | 6.89 ↑ | + 0.40 | + 0.15 |

↑ Better than the baseline.  ↓ Worse than the baseline.

## 4.1 PRELIMINARY STUDY OF FEATURE SELF-CALIBRATION OPERATIONS

We conducted a controlled experiment to test the introduced self-calibration operations on CIFAR-10 and CIFAR-100. Specifically, for the proposed *dynamic convolutions*, we considered both normal and dilated convolutions and two kernel sizes ($3 \times 3$ and $5 \times 5$). We adopted the channel-wise and spatial-wise *attention learning*. For the baseline model, we used ResNet-18 (He et al., 2016) with 4 stages in the backbone. To build a model with self-calibration, we added each self-calibration operation at the stages 1, 2, 3 of ResNet-18, respectively. For fair comparison, we trained each model in the same setting (see Appendix A.2.1). In Table 1, we summarised the model parameters and FLOPs in addition to the test set performance (error rates). We observed that: (1) Both attention operations and our normal dynamic convolutions outperform the baseline consistently; (2) Adding dilated dynamic convolutions causes performance drop in most cases. We hence exclude it from the candidate set; (3) Very marginal FLOPs and parameters increase from these self-calibration operations over the baseline, suggesting their high cost-effectiveness.

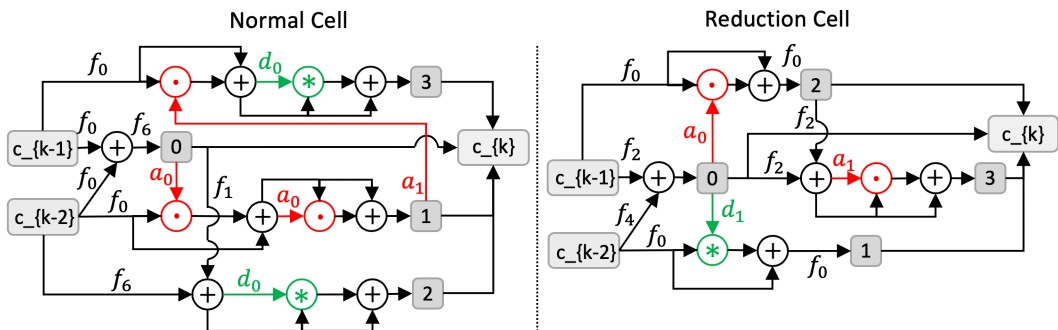

Figure 4: Normal cell and reduction cell searched on CIFAR-10. $f_0$: sep_conv_3x3, $f_1$: sep_conv_5x5, $f_2$: dil_conv_3x3, $f_4$: max_pooling, $f_6$: identity, $a_0$: spatial_attention, $a_1$: channel_attention, $d_0$: dynamic_conv_3x3, $d_1$: dynamic_conv_5x5.

## 4.2 CELL SEARCH

**Search Space.** As found out above, the heterogeneous primitive operation set $\mathcal{O}$ contains 11 operations in total: $|\mathcal{O}_f| = 7$ feature transformation operations, $|\mathcal{O}_a| = 2$ attention learning operations, $|\mathcal{O}_d| = 2$ dynamic convolutions, respectively. We constructed the proposed heterogeneous operator cell ($\mathcal{G} = (\mathcal{V}, \mathcal{E})$) with $|\mathcal{V}| = 7$ nodes (2 input nodes, 4 inner nodes and 1 output node). So, all 4 heterogeneous residual blocks contain $|\mathcal{E}| = 18$ edges in total (14 first-tier connections and 4 second-tier connections). To derive the final cell architecture, we kept 2 first-tier connections and 1 second-tier connection for each block. As a result, there is a total number of $\prod_{n=1}^{4} \frac{(n+1)n}{2} \times 11^3 \approx 10^{14}$ possible choices, 5 orders of magnitude larger than the conventional size of $\prod_{n=1}^{4} \frac{(n+1)n}{2} \times 7^2 \approx 10^9$ as in (Liu et al., 2019b; Dong & Yang, 2019; Xie et al., 2019).

**Training.** Following the setup of existing methods (Real et al., 2019; Liu et al., 2019b; 2018a; Akimoto et al., 2019), we searched the convolutional architectures on CIFAR10. We constructed a

Table 2: Comparisons with the state-of-the-art architectures on CIFAR10 and CIFAR100.

| Architecture | Error (%) | | Params | Search Cost | | Type |
|---|---|---|---|---|---|---|
| | CIFAR10 | CIFAR100 | (M) | GPUs | Days | |
| PyramidNet (Han et al., 2017)[*] | 3.92 | 20.11 | 2.5 | - | - | Manual |
| DenseNet-BC (Huang et al., 2017) | 3.46 | 17.18 | 25.6 | - | - | Manual |
| NASNet-A (Zoph et al., 2018) | 2.65 | - | 3.3 | 450 | 1800 | RL |
| AmoebaNet-B (Real et al., 2019) | 2.55±0.05 | - | 2.8 | 450 | 3150 | EA |
| Hierarchical-Evolution (Liu et al., 2018b) | 3.75±0.12 | - | 15.7 | 200 | 300 | EA |
| PNAS (Liu et al., 2018a) | 3.41±0.09 | - | 3.2 | 100 | 1.5 | SMBO |
| ENAS (Pham et al., 2018) | 2.89 | - | 4.6 | 1 | 0.5 | RL |
| ProxylessNAS Cai et al. (2019) | **2.08** | - | 5.7 | - | 4 | GD |
| RENAS (Chen et al., 2019) | 2.88±0.02 | - | 3.5 | 4 | 6 | EA&RL |
| DARTS(1st) (Liu et al., 2019b) | 3.00±0.14 | - | 3.3 | 1 | 1.5 | GD |
| DARTS(2nd) (Liu et al., 2019b) | 2.76±0.09 | 17.54 | 3.3 | 1 | 4.0 | GD |
| SNAS (moderate) (Xie et al., 2019) | 2.85±0.02 | - | 2.8 | 1 | 1.5 | GD |
| GHN (Zhang et al., 2019) | 2.84±0.07 | - | 5.7 | 1 | 0.84 | GD |
| GDAS (Dong & Yang, 2019) | 2.93 | 18.38 | 3.4 | 1 | 0.84 | GD |
| BayesNAS(0.005) (Zhou et al., 2019) | 2.81±0.04 | - | 3.4 | 1 | 0.2 | GD |
| ASNG (Akimoto et al., 2019) | 2.83±0.14 | - | 3.9 | 1 | **0.11** | GD |
| Random Baseline[‡] | 3.85 | 21.66 | 2.4 | - | - | Random |
| **NOS (best)** | 2.53 | **16.21** | 2.6 | 1 | 0.35 | GD |
| **NOS (average)** | 2.67±0.06 | **16.72±0.24** | 2.6 | 1 | 0.35 | GD |

[*] The teacher model.   [‡] Best architecture among 30 random samples.

small proxy network with 8 heterogeneous operator cells, and two reduction cells at 1/3 and 2/3 of the total network depth for feature shape reduction. We used 25K images split from the training set for validation. We randomly initialised the architecture parameters $A \in \mathbb{R}^{18 \times 11}$ in the normal distribution. We used a pre-trained PyramidNet-110 (bottleneck, $\alpha = 84$) (Han et al., 2017) as the teacher model. We set the weight $\lambda = 10^3$ for attention guidance loss $\mathcal{L}_{AT}$. After 25 epochs of training on the proxy network, we derived the final heterogeneous operator cells from the architecture matrix $A$. See Appendix A.2.1 for more configurations for training the proxy and teacher networks.

The search on CIFAR10 took only 8.4 hours using a single NVIDIA Tesla V100 GPU. The searched heterogeneous operator cells by NOS is shown in Fig. 4, in which the self-calibration operators $\odot$ and $\circledast$ appear in both first-tier and second-tier. For example, there are two *attention* operations in first-tier and two *dynamic convolutions* in second-tier in the normal cell.

### 4.3   ARCHITECTURE EVALUATION

**CIFAR.** To measure the final image classification performance of the searched heterogeneous operator cells on CIFAR10 and CIFAR100, we created an evaluation network with 20 cells, 36 initial channels, and an auxiliary tower with loss weight 0.4. See Appendix A.2.1 for more configurations for training the evaluation network. Due to high variance of results on CIFAR, we conducted 10 independent runs and reported both the best and average results. We summarised the results of NOS and the state-of-the-art methods in Table 2. The comparisons show that: **(1)** NOS achieves a very competitive result (second best) on CIFAR10, whilst enjoying the smallest model parameters (only 2.6M). Comparing with the best performer ProxylessNAS at the size of 5.7M (searched with a *supernet*) (Cai et al., 2019), it shows the significant cost-effectiveness and compactness advantages of our method. **(2)** Despite a significantly larger search space ($10^{14}$ vs $10^9$ in (Liu et al., 2019b; Akimoto et al., 2019; Xie et al., 2019; Dong & Yang, 2019; Akimoto et al., 2019)), NOS shows high cost-effectiveness in computing cost (only 0.35 GPU day). **(3)** NOS achieves the best result on CIFAR100 by directly transferring the CIFAR10 searched network, significantly outperforming DARTS (Liu et al., 2019b) and GDAS (Dong & Yang, 2019). This challenging cross-dataset test indicates a superior transferability of the network searched by NOS.

**ImageNet.** To evaluate the transferability of architecture discovered by NOS on large scale ImageNet, we used the mobile setting same as in (Liu et al., 2019b; Dong & Yang, 2019; Xie et al., 2019), where the number of multiply-add operations is restricted to be less than 600M at the input

Table 3: Comparisons with the state-of-the-art architectures on ImageNet-mobile.

| Architecture | Test Err. (%) | | Params | $\times+$ | Search Cost | Type |
|---|---|---|---|---|---|---|
| | top-1 | top-5 | (M) | (M) | (GPU-days) | |
| MobileNet-v1(1.0) (Howard et al., 2017) | 29.4 | 10.5 | 4.2 | 575 | - | Manual |
| MobileNet-v2(1.0) (Sandler et al., 2018) | 28.0 | - | 3.4 | 300 | - | Manual |
| ShuffleNet 2×(v1) (Zhang et al., 2018) | 26.4 | 10.2 | ≈5 | 524 | - | Manual |
| ShuffleNet 2×(v2) (Ma et al., 2018) | 25.1 | - | ≈5 | 591 | - | Manual |
| NASNet-A (Zoph et al., 2018) | 26.0 | 8.4 | 5.3 | 564 | 1800 | RL |
| NASNet-B (Zoph et al., 2018) | 27.2 | 8.7 | 5.3 | 488 | 1800 | RL |
| NASNet-C (Zoph et al., 2018) | 27.5 | 9.0 | 4.9 | 558 | 1800 | RL |
| PNAS (Liu et al., 2018a) | 25.8 | 8.1 | 5.1 | 588 | 1.5 | SMBO |
| AmoebaNet-A (Real et al., 2019) | 25.5 | 8.0 | 5.1 | 555 | 3150 | EA |
| AmoebaNet-B (Real et al., 2019) | 26.0 | 8.5 | 5.3 | 555 | 3150 | EA |
| AmoebaNet-C (Real et al., 2019) | 24.3 | 7.6 | 6.4 | 570 | 3150 | EA |
| RENAS (Chen et al., 2019) | 24.3 | 7.4 | 5.4 | 580 | 6 | EA&RL |
| MnasNet-A3 (Tan et al., 2019) | **23.3** | **6.7** | 5.2 | 403 | -[†] | RL |
| ProxylessNAS (GPU) (Cai et al., 2019) | 24.9 | 7.5 | 7.1 | 465 | 8.3 | GD |
| FBNet-C (Wu et al., 2019a) | 25.1 | - | 5.5 | **375** | 9.0 | GD |
| GHN (Zhang et al., 2019) | 27.0 | 8.7 | 6.1 | 569 | 0.84 | GD |
| DARTS (Liu et al., 2019b) | 26.7 | 8.7 | 4.7 | 574 | 4.0 | GD |
| SNAS (Xie et al., 2019) | 27.3 | 9.2 | 4.3 | 522 | 1.5 | GD |
| GDAS (Dong & Yang, 2019) | 26.0 | 8.5 | 5.3 | 581 | 0.84 | GD |
| BayesNAS (0.005) (Zhou et al., 2019) | 26.5 | 8.9 | 3.9 | - | 0.2 | GD |
| **NOS (searched on CIFAR10)** | 25.8 | 8.1 | 4.0 | 440 | 0.35 | GD |

[†] The architecture search takes 4.5 days on 64 TPUv2 devices.

size of $224 \times 224$. Specifically, we constructed an evaluation network with 14 cells and 48 initial channels. An auxiliary tower with loss weight 0.4 was also applied. See Appendix A.2.2 for more training details. Table 3 shows the ImageNet results in the mobile setting. Notably, the cell architectures found by NOS on CIFAR10 can achieve highly competitive performance with significantly less computational cost (0.35 day on 1 GPU vs 4.5 days using 64 TPUv2 devices required by MnasNet-A3 (Tan et al., 2019)). Unlike MnasNet-A3 (Tan et al., 2019) and ProxylessNAS (Cai et al., 2019) searching the network on ImageNet directly (resource-intensive), the network searched by NOS on CIFAR10 can be successfully transferred. Also, compared to other state-of-the-art gradient based proxy-based NAS (GHN, DARTS, SNAS, GDAS and BayesNAS), NOS discovers a cell structure that performs better with higher efficiency (only 440M FLOPs).

## 4.4 FURTHER ANALYSIS

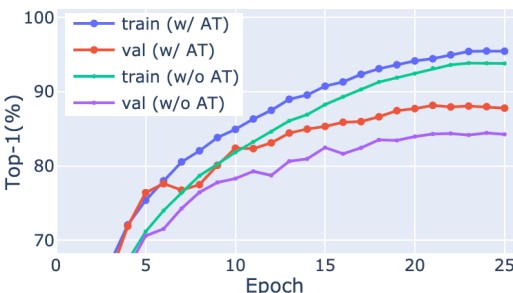

Figure 5: Train/Val set accuracy on CIFAR10.

Table 4: Testing attention guided search (AGS).

| AGS | Test Error (%) | |
|---|---|---|
| | CIFAR10 | CIFAR100 |
| w/o | 3.44 | 18.80 |
| w/ | 2.53 | 16.21 |

We evaluated attention guided search (AGS) on CIFAR10/100 by comparing a NOS variant without attention transfer loss. The same training setting was used (Appendix A.2.1)[1]. We used a pre-trained PyramidNet-110 as teacher. Table 4 shows that learning with attention guidance can significantly benefit the NOS search process. We further showed the training curves in Fig. 5 and observed that

---

[1]The absence of AGS gives a slight difference to the search cost: GPU day 0.34 vs. 0.35 when using AGS.

AGS clearly improves the train/val accuracies. This suggests that AGS is effective to alleviate the architecture training-test discrepancy issue involved in the proxy-based NAS.

## 5 CONCLUSION

We presented Neural Operator Search (NOS), featured by a heterogeneous search space for neural architecture search (NAS). This search space expansion enables NAS to discover more expressive and previously undiscovered architectures, significantly expanding the search horizon and enriching the possible search outcomes. We further formulated heterogeneous residual block and attention guided search to solve the intrinsic search challenges involved. Extensive experiments on image classification show that NOS can discover novel and high-quality cell architectures in a cost-effective process. We hope that this work will shed light on the future directions for the NAS community.

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

## A   APPENDIX

### A.1   SELF-CALIBRATION OPERATIONS

#### A.1.1   ATTENTION LEARNING

```python
import torch
import torch.nn as nn
import torch.nn.functional as F

# channel-wise attention
class AttentionC(nn.Module):

  def __init__(self, C_in, C_out, reduction=16, affine=True):
    super(AttentionC, self).__init__()
    self.conv_1 = nn.Conv2d(C_in, C_in // reduction, 1, stride=1, padding=0, bias=False)
    self.relu = nn.ReLU(inplace=False)
    self.conv_2 = nn.Conv2d(C_in // reduction, C_out, 1, stride=1, padding=0, bias=False)
    self.sigm = nn.Sigmoid()

  def forward(self, x):

    y = F.avg_pool2d(x, kernel_size=x.size()[2:4])
    y = self.relu(self.conv_1(y))
```

```
20    y = self.sigm(self.conv_2(y))
21    return y
22
23  # spatial-wise attention
24  class AttentionS(nn.Module):
25
26    def __init__(self, C_in, C_out, stride, reduction=16, affine=True):
27      super(AttentionS, self).__init__()
28
29      self.conv_1 = nn.Conv2d(C_in, C_in // reduction, 1, stride=stride,
        padding=0, bias=False)
30      self.bn_1 = nn.BatchNorm2d(C_in // reduction, affine=affine)
31      self.relu = nn.ReLU(inplace=False)
32      self.conv_2 = nn.Conv2d(C_in // reduction, 1, 3, stride=1, padding=1,
         bias=False)
33      self.sigm = nn.Sigmoid()
34
35    def forward(self, x):
36
37      y = self.relu(self.bn_1(self.conv_1(x)))
38      y = self.sigm(self.conv_2(y))
39      return y
```

### A.1.2 DYNAMIC CONVOLUTIONS

```
1  # dynamic convolution
2  class DynamicF(nn.Module):
3
4    def __init__(self, C_in, C_out, F_size, reduction=8, affine=True):
5      super(DynamicF, self).__init__()
6      self.f_size = F_size
7      self.reduction = reduction
8      self.c_in = C_in
9      self.c_out = C_out
10      self.conv_1 = nn.Conv2d(C_in, C_in // reduction, 1, stride=1, padding
        =0, bias=False)
11      self.bn_1 = nn.BatchNorm2d(C_in // reduction, affine=affine)
12      self.relu = nn.ReLU(inplace=False)
13      self.conv_2 = nn.Conv2d(C_in, F_size*F_size, 1, stride=1, padding=0,
        bias=False)
14      self.soft = nn.Softmax(dim=2)
15      self.conv_3 = nn.Conv2d(C_in // reduction, C_out, 1, stride=1,
        padding=0, bias=False)
16
17    def forward(self, x):
18
19      input_x = self.relu(self.bn_1(self.conv_1(x)))
20      N, C, H, W = input_x.size()
21      input_x = input_x.view(N, C, H * W)               # [N, C, H * W]
22      input_x = input_x.unsqueeze(1)                    # [N, 1, C, H * W]
23      dynamic_mask = self.conv_2(x)                     # [N, -1, H , W]
24      dynamic_mask = dynamic_mask.view(N, -1, H * W)    # [N, -1, H * W]
25      dynamic_mask = self.soft(dynamic_mask)            # [N, -1, H * W]
26      dynamic_mask = dynamic_mask.unsqueeze(3)          # [N, -1, H * W, 1]
27      dynamic_mask = dynamic_mask.permute(0, 3, 2, 1)   # [N, 1, H * W, -1]
28      dynamic = torch.matmul(input_x, dynamic_mask)     # [N, 1, C, -1]
29      dynamic = dynamic.squeeze(1)                      # [N, C, -1]
30      dynamic = dynamic.view(N, C, self.f_size, self.f_size)
31      dynamic = self.conv_3(dynamic)
32      dynamic = dynamic.view(N, self.c_out, -1)
33      dynamic = self.soft(dynamic)
34      dynamic = dynamic.view(N, self.c_out, self.f_size, self.f_size)
35      return dynamic
```

## A.2 DETAILS OF TRAINING CONFIGURATIONS

### A.2.1 CIFAR

**ResNet-18 and PyramidNet-110.** We trained these models for 300 epochs with batch size 32. The learning rate was initialised as 0.025, which was decayed by 10 every 30 epochs. The standard SGD optimiser with momentum of 0.9 was employed. We set a weight decay value of $1 \times 10^{-4}$ to avoid overfitting. Other additional enhancements were not involved except the standard data augmentations.

**Cell Search.** For network parameters $\Theta$ of proxy network, we used SGD with an initial learning rate 0.025 and set the momentum value as 0.9. This learning rate was decayed to 0 with a cosine scheduler. A weight decay value of $3 \times 10^{-4}$ was imposed to avoid over-fitting. For learning architecture matrix $A$, we used the Adam optimiser with a fixed learning rate value $6 \times 10^{-4}$ and set the weight decay to $1 \times 10^{-3}$.

**Cell Evaluation.** The evaluation network was trained from scratch directly for 600 epochs with batch size 128. Note that, the attention transfer was **not** involved for training. We set the weight decay values for CIFAR-10 and CIFAR-100 to $3 \times 10^{-4}$ and $5 \times 10^{-4}$ individually. The standard SGD optimiser with a momentum of 0.9 was applied. The initial learning rate was 0.25, decayed to 0 with a cosine scheduler. Following existing works (Liu et al., 2019b; Pham et al., 2018; Zoph et al., 2018; Real et al., 2019), we performed two additional enhancements: the cutout regularisation (DeVries & Taylor, 2017) with length 16 and the drop-path (Larsson et al., 2017) of probability 0.3.

### A.2.2 IMAGENET

We trained the evaluation model for ImageNet using SGD optimiser for 300 epochs with batch size 512. We initialised the learning rate as 0.25 and reduced it to 0 by a linear scheduler. Learning rate warmup (Goyal et al., 2017) was applied for the first 5 epochs to deal with the large batch size and learning rate.

## A.3 VISUALISATION OF SEARCH SPACE

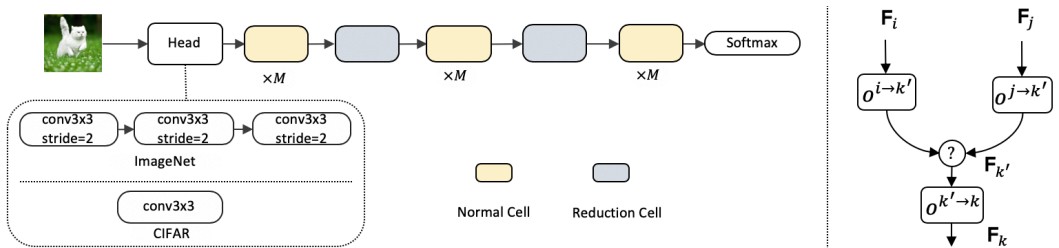

Figure 6: (**Left**) The overall model architecture for CIFAR-10 and ImageNet, consisting of repeated Normal Cells and Reduction Cells. $M$ is the stacking choice for the number of Normal Cells. Each cell contains 4 blocks. (**Right**) An example of two-tier block construction in cell: Each block takes two input features ($\mathbf{F}_i$, $\mathbf{F}_j$) from previous nodes; The operator (?) in a block is determined by the choices of two operations ($o^{i \to k'}$, $o^{j \to k'}$) in the first-tier; An extra operation ($o^{k' \to k}$) is selected in the second-tier.

## A.4 DISTILLATION EFFECT IN ATTENTION GUIDED SEARCH

We examined the effect of knowledge distillation in the proposed Attention Guided Search (AGS). We conducted this analysis on CIFAR10. In this evaluation, we compared three methods: (1) Vanilla Search: Using the original DARTS search method; (2) Distillation Only: Using the attention transfer loss for training the proxy network only; (3) AGS: The proposed method (full). We tracked the model performance on both training (train) and validation (val) data sets. Figure 7 shows that (i)

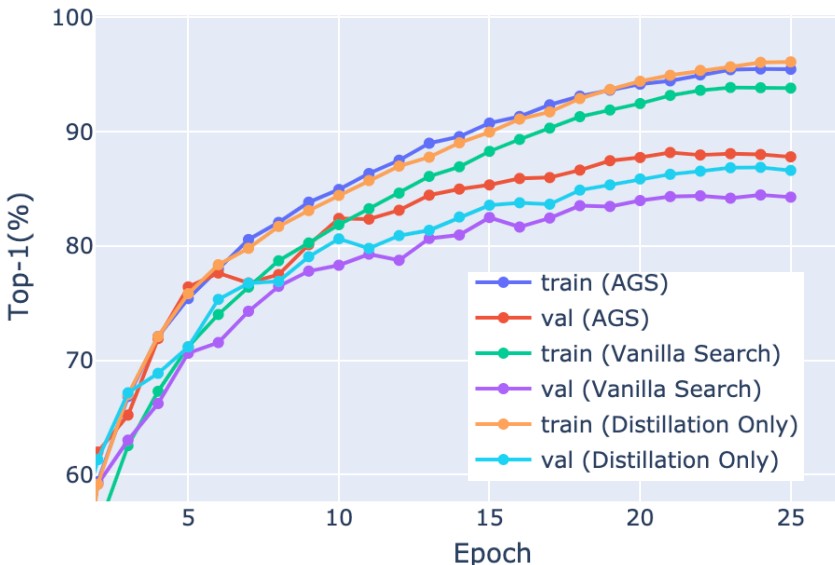

Figure 7: The train and val set accuracies on CIFAR10 in different search processes.

knowledge distillation brings a positive performance gain over the vanilla DARTS and (ii) using attention guidance for the network search can further improve the searched architecture.

## A.5 GENERALITY OF ATTENTION GUIDED SEARCH

We tested the general effect of the proposed Attention Guided Search (AGS) using the original DARTS search space ($\mathcal{O}_f$ + zero operation) on CIFAR10. During the search process, we followed the same settings as DARTS with the first-order optimisation. We obtained the error rates: 3.00 $\pm$ 0.14 (DARTS) vs. 2.92 $\pm$ 0.05 (DARTS + AGS). This suggests a general efficacy of AGS over different search spaces.

