# OpenReview forum: "Neural Operator Search"
_ICLR.cc/2020/Conference — Reject_

### Official Review · AnonReviewer1 · 2019-10-17
**Official Blind Review #1**

**Rating:** 6

**Review:**

Summary:

Often in (neural architecture search) NAS papers on vision datasets, only feature transform operations (e.g. convolution, pooling, etc) are considered while operations like attention and dynamic convolution are left out. These operations have shown increasing performance improvements across many domains and datasets. This paper attempts to incorporate these operations into the search space of NAS.

Specifically they design a new cell (residual block in a resnet backbone) which contains these operations (traditionally used feature transforms, attention, and dynamic convolutions). This new cell contains two-tiers. The first tier separately computes the three kinds of operations and combines the results via elementwise summation to form an 'intermediate calibrated tensor'. This is then fed to the second tier where again the three kinds of operations are performed on them and they net output is again formed via elementwise summation.

For the search algorithm the authors use the bilevel optimization of DARTS. The one difference to aid in the search is that an 'attention-guided' transfer mechanism is used where a teacher network trained on the larger dataset (like ImageNet) is used to align the intermediate activations of the proxy network found during search, by adding a alignment loss function.

Results of search with this new search space on cifar10 and transfer to cifar100, imagenet are presented. It is found via manual ablation in resnet backbone that dilated dynamic convolutions dont help and are dropped from the search space.

The numerical results are near state-of-the-art although as is pervasive in the NAS field actual fair comparisons between methods are hard to get due to differences in search space, hardware space, stochasticity in training etc. But the authors do a best-attempt and have included random search and best and average performances so that is good.

Comments:

- The paper is generally well-written and easy to understand (thanks!)

- Experiments are generally thorough.

- The main novelty is the design of the cell such that attention and dynamic convolution operations can be incorporated. I was hoping that the results in terms of error would be much better due to the more expressive search space but they are not at the moment.

- I am curious how the attention guided alignment loss has been ablated. How do the numbers look without alignment loss keeping gpu training time constant? Basically I am trying to figure out how much is the contribution of the new search space vs. the addition of attention guided search. Do we have a negative result (an important one though) that attention and dynamic convolutions don't really help?


**Experience Assessment:**

I have published one or two papers in this area.

**Review Assessment: Checking Correctness Of Derivations And Theory:**

N/A

**Review Assessment: Checking Correctness Of Experiments:**

I carefully checked the experiments.

**Review Assessment: Thoroughness In Paper Reading:**

I read the paper thoroughly.

---

> ### Author Response · Authors · 2019-11-10
> **Response to Blind Review #1**
>
> We would like to thank Reviewer #1 for the constructive comments and positive recommendation.
> Our responses inline:
>
> Comment 1: “I was hoping that the results in terms of error would be much better due to the more expressive search space, but they are not at the moment.”
>
> This shares our very initial expectation which also underpins the core motivation of this work. However, our elaborative investigations suggest that it is non-trivial to achieve super-expressive results from using a more expressive search space. This reason is that such a richer and heterogeneous search space imposes more significant challenges to the search process due to being exponentially larger and an extra need for integrating different types of tensor operations coherently. Whilst solving the NAS problem is inherently more challenging, it also promises enormous potential for discovering superior network architectures, which are previously unexplored due to the limitation of search space. This potential (see experiments) has been notably demonstrated in this very first attempt with two key components: a heterogeneous residual block and the attention guided search. And we hope more follow-up works can further advance this direction to find out more advanced network structures. For example, searching the whole network on a target dataset (e.g. ImageNet) directly in a cost-effective manner. We believe this work opens a new exciting territory for the NAS community.
>
>
> Comment 2: “I am curious how the attention guided alignment loss has been ablated. How do the numbers look without alignment loss keeping GPU training time constant?”
>
> For ablation study of attention guided search, we simply train the proxy network w/ attention transfer loss and w/o attention transfer loss, while keeping all the others (including the epoch number) unchanged, for a direct and fair comparison. The training time for the two experiments is different: GPU day 0.35 (w/) vs 0.34 (w/o). We have clarified this in the manuscript (see footnote 1 on Page 9).
>
>
> Comment 3: “I am trying to figure out how much is the contribution of the new search space vs. the addition of attention guided search.”
>
> First of all, we note that attention guided search is part of our search algorithm based on DARTS first-order gradient optimisation, designed specifically for aiding the searching process in an exponentially larger space. As shown in Table 4, with the vanilla DARTS, it is much harder to search over this newly proposed search space as compared to the standard search space (please also refer to the response to Comment 1 above). This suggests that different search spaces should be exploited by corresponding different search algorithms in order to well mine their intrinsic potentials. It is, therefore, improper to evaluate the effect of a search space and a search algorithm separately.
>
> Besides, as suggested by Reviewer #3, we have additionally performed attention guided search over the original DARTS search space and shown a positive impact. Please see the response to Comments 4 and 5 of Reviewer #3 and Appendix 4 & 5.
>
>
> Comment 4: “Do we have a negative result (an important one though) that attention and dynamic convolutions don't really help?”
>
> Yes. As shown in Table 4, without attention transfer loss to aid the search process, the searched architectures, containing both multiple attention operations and dynamic convolutions, turn out to perform not well on CIFAR-10/100 datasets. Again, improving the search space leads to a need for refining the search algorithm accordingly.

---

> > ### Comment · AnonReviewer1 · 2019-11-11
> > **Response acknowledged!**
> >
> > Thanks for the ablation of AGS and the new results on DARTS search space.
> >
> > I also agree that extracting performant architectures from more expressive search space is not trivial. It is good that attention and dynamic convolutions helps existing architectures so there is hope.
> >
> > Would the authors agree that the main contribution of this paper at the moment is the new expressive search space with attention etc and showing that this is promising?

---

> > > ### Author Response · Authors · 2019-11-11
> > > **About the main contribution**
> > >
> > > Thanks for your response.
> > >
> > > Overall, yes, this study demonstrates that a search space with dynamic convolution and attention is promising.
> > >
> > > However, this is not as trivial as just simply introducing these operations into the primitive operator set, due to that a heterogeneous search space is resulted, vs. the conventional homogeneous search space with only feature transformation operators. We overcome this extra challenge by designing a novel Heterogeneous Residual Block (Figure 2), which enables us to exploit existing state-of-the-art search algorithms. This, we consider, is a significant contribution, beyond adding extra operators.
> > >
> > > Another non-trivial contribution is that we introduce a novel dynamic convolution structure (Figure 1) tailored for image representation learning, with both efficiency and effectiveness in mind. We argue it has an elegant holistic design.
> > >
> > > In terms of the search algorithm, we demonstrate that attention guided search adds values to DARTS (a prior art method), particularly in the newly proposed search space. Our extra experiment shows that this idea can also benefit the existing methods (Appendix 5). This is generally interesting to know with significance.
> > >
> > > In summary, this work not only contributes a richer NAS search space, but also offers a neat NAS solution established upon a set of novel and elegant designs.

---

### Official Review · AnonReviewer3 · 2019-10-23
**Official Blind Review #3**

**Rating:** 3

**Review:**

This paper proposes a complicated NAS approach, with a lot more ops, a larger search space and an extra teacher network. However, the results on CIFAR10 and ImageNet are both not competitive to other SOTA results.

My major concern is whether the extra complexity introduced by this paper is necessary:

1. The new search space includes a lot more ops and optimizations, which by themselves should improve other models. For example, according to Table1, dynamic conv and attention improve ResNet-18 without any architecture search. What if you simply apply the same optimizations to other SOTA models in Table 2 and 3?
2. The extra ops already make the comparison in Table 2/3 unfair. Despite that, NOS is still not competitive to other SOTA results (prroxylessNAS on CIFAR-10 and MnasNet-A3 on ImageNet).
3. The authors argue that “NOS shows significant compactness advantages than proxylessNAS”, but it is somewhat misleading. By reading this paper, it seems the search algorithm used in this paper aims to find the highest accuracy model WITHOUT resource constraints, so smaller model size should not be related to the search algorithm.

Here are a few other comments and suggestions:

4. Section 3.2 is difficult to follow. I recommend the authors providing some visualization for the search space (see NASNet paper for example).
5. Ablation study is relatively weak: for example, Figure 5 compares w/ and w/o attention-guided search, but it is unclear whether the gain is by the search or by the teacher distillation. A more fair comparison is to w/o attention-guided search but also perform the distillation. It would be also helpful to verify whether the propose attention-guided search can work for existing  search space (such as the NASNet/DARTS search space).

**Experience Assessment:**

I have published in this field for several years.

**Review Assessment: Checking Correctness Of Derivations And Theory:**

I assessed the sensibility of the derivations and theory.

**Review Assessment: Checking Correctness Of Experiments:**

I assessed the sensibility of the experiments.

**Review Assessment: Thoroughness In Paper Reading:**

I read the paper at least twice and used my best judgement in assessing the paper.

---

> ### Author Response · Authors · 2019-11-10
> **Response to Blind Review #3**
>
> We would like to thank Reviewer #3 for the constructive comments. Our responses inline:
>
> Comment 1: The new search space includes a lot more ops and optimizations, which by themselves should improve other models. For example, according to Table1, dynamic conv and attention improve ResNet-18 without any architecture search. What if you simply apply the same optimizations to other SOTA models in Table 2 and 3?
>
> As we did for ResNet18 in Section 4.1, this was also attempted in the first version of MnasNet (Tan et al., https://arxiv.org/pdf/1807.11626v1.pdf, see Table 1 in Section 4). Specifically, Tan et al. show that further adding the SE-module (i.e. the channel-wise attention) to a pre-searched model boosts the performance (consistent with our findings in Table 1 in Section 4, see the discussion in “Self-calibration” in Section 2). We hope these evidences and attempts can address the main concern of the reviewer on “whether the extra complexity introduced by this paper is necessary” because they are useful for improving the model generalisation. Beyond hand-design of dynamic convolution and attention learning, in this work, we further investigate them into the NAS context, with the aim that their potentials can be better mined through automatic structure search. This enables more sophisticated and advanced architectures to be discovered in NAS, which has great potentials.
>
>
> Comment 2: The extra ops already make the comparison in Table 2/3 unfair. Despite that, NOS is still not competitive to other SOTA results (ProxylessNAS on CIFAR-10 and MnasNet-A3 on ImageNet). The authors argue that “NOS shows significant compactness advantages than proxylessNAS”, but it is somewhat misleading. By reading this paper, it seems the search algorithm used in this paper aims to find the highest accuracy model WITHOUT resource constraints, so smaller model size should not be related to the search algorithm.
>
> First of all, we argue that this opinion of unfair comparison due to extra operations is largely biased. In this viewpoint, it is implied that many existing influential works were making and based on “unfair comparisons”, e.g. attention models vs. non-attention models, dynamic convolution vs. static convolution, residual networks vs. non-residual networks, proxyNAS vs proxyless NAS, etc. This clearly cannot stand. In general, research innovations often lead to new notions/concepts (out of the existing ones) in order to make breakthrough solutions beyond the previous state-of-the-art.
> As acknowledge by Reviewer #1, it is frustratingly hard (if possible) to achieve fair comparisons, due to different search spaces, different accessible computational resources (GPUs or TPUs), different training strategies, and etc. used in the existing NAS studies. Concretely, in terms of the search space, ProxylessNAS [1] uses a tree-structured architecture space based on PyramidNet [2], and MnasNet-A3 added a pre-designed convolution layer with a SE-module (i.e. the channel-wise attention). In terms of the computational cost, ProxylessNAS performs the search with 4 GPU days vs 0.35 GPU day in our NOS, and MnasNet [3] benefits from the powerful 64 TPUv2 devices which makes the direct search on ImageNet in 4.5 days possible. In contrast, our architectures for ImageNet are searched on CIFAR-10 due to the resource constraint. Such differences prevent a direct and fully fair comparison between different works; Consequently evaluating a NAS work only from the end performance is intrinsically biased (This is why we compared our NOS with ProxylessNAS in both searching cost and final model size). In short, we have done our best attempts in the standard evaluation criteria as we can. We welcome any specific suggestions for further improving this work in our resource limit.
>
> [1] Cai et.al. Proxylessnas: Direct neural architecture search on target task and hardware. In ICLR, 2019.
> [2] Han et.al. Deep pyramidal residual networks. In CVPR, 2017.
> [3] Tan et.al. Mnasnet: Platform-aware neural architecture search for mobile. In CVPR, 2019.
>
>
> Comment 3: Section 3.2 is difficult to follow. I recommend the authors providing some visualization for the search space (see NASNet paper for example).
>
> Many thanks. As suggested, we have improved our manuscript with additional visualisation for the search space. Please see Figure 6 in Appendix 3.

---

> > ### Author Response · Authors · 2019-11-10
> > **Continue**
> >
> >
> > Comment 4: [Ablation study] Figure 5 compares w/ and w/o attention-guided search, but it is unclear whether the gain is by the search or by the teacher distillation. A more fair comparison is to w/o attention-guided search but also perform the distillation.
> >
> > A good point. We assume this comment means that, the distillation is only applied to training the proxy network (Eq. (8)), but not to the architecture search (Eq. (9)). We have additionally conducted this evaluation. Overall, we found that distillation helps find a better network architecture and attention guided search further improves. Please see Appendix 4.
> >
> > Comment 5: [Ablation study] It would be also helpful to verify whether the propose attention-guided search can work for existing search space (such as the NASNet/DARTS search space).
> >
> > Good suggestion. We have now additionally tested the proposed Attention Guided Search (AGS) using the original DARTS [1] search space (O_f + zero operation, i.e. feature learning) on CIFAR-10. During the search process, we followed the same settings as DARTS with the first-order optimisation. We obtained the error rates:
> > DARTS: 3.00 ± 0.14
> > DARTS + AGS: 2.92 ± 0.05
> > This suggests a general efficacy of AGS over different search spaces. We have added this evaluation in Appendix 5.
> >
> > [1] Liu et.al. Darts: Differentiable architecture search. In ICLR, 2019.

---

### Official Review · AnonReviewer2 · 2019-10-23
**Official Blind Review #2**

**Rating:** 3

**Review:**

This paper proposes a new method for neural architecture search that searches architectures with not only feature transformational components but also self-calibration and dynamic convolution ones. Although the idea seems to be straightforward and innovating, it is difficult to assess the effectiveness of the proposed approach comparing to other methods just by looking at the experimental results.

Looking at the CIFAR10 results, the authors claimed that their method achieved second best among all the method compared but with significantly less number of parameters. Although indeed the number of parameters differ by a lot, the error rates are also differed significantly. It might be worth comparing these two baselines using the same number of parameters just to conduct a fair comparison.

In the Imagenet results, the proposed method is behind many of the state-of-the-art methods, casting concerns on the effectiveness of the proposed approach considering its added search space.

**Experience Assessment:**

I have published one or two papers in this area.

**Review Assessment: Checking Correctness Of Derivations And Theory:**

I did not assess the derivations or theory.

**Review Assessment: Checking Correctness Of Experiments:**

I carefully checked the experiments.

**Review Assessment: Thoroughness In Paper Reading:**

I made a quick assessment of this paper.

---

> ### Author Response · Authors · 2019-11-10
> **Response to Blind Review #2**
>
> We would like to thank Reviewer #2 for the constructive comments. Our responses inline:
>
> Comment 1: “Looking at the CIFAR10 results, the authors claimed that their method achieved second-best among all the method compared but with significantly less number of parameters. Although indeed the number of parameters differ by a lot, the error rates also differ significantly. It might be worth comparing these two baselines using the same number of parameters just to conduct a fair comparison.”
>
> First of all, it should be noted that we did follow the standard NAS evaluation protocol, without introducing any new comparison criteria. In general, model accuracy and model size are the two most important evaluation metrics. To our best knowledge, it is very rare (if any) that the NAS studies compare different network architectures with subject to the same parameter numbers. Three reasons make this practically infeasible: (1) The final networks obtained by different NAS methods often vary significantly in the model parameter size; (2) It is non-trivial to accurately control the parameter number of a final network architecture in NAS; (3) Competitors are often of different parameter sizes, which renders the exhaustive comparisons prohibitively expensive.
>
>
> Comment 2: “In the Imagenet results, the proposed method is behind many of the state-of-the-art methods, casting concerns on the effectiveness of the proposed approach considering its added search space.”
>
> In NAS, it is generally hard to do a direct and fully fair comparison (as agreed by Reviewer #1), due to the differences in the search spaces, the computational resources, the training strategies, etc. While performance is important, it is NOT all of the research work. In our context, the best competitor of our method should be DARTS [1] since we focus on (1) the search space improvement but not the search algorithm and (2) we used DRATS as the base search algorithm. Please see our response to Comment 2 of Reviewer #3.
>
>
> Comment 3: “It is difficult to assess the effectiveness of the proposed approach comparing to other methods just by looking at the experimental results”.
>
> This comment is too abstract to let us act to further improve this work. Overall, we argue that the evaluations have been thorough and sufficient for efficacy assessment, along with clearly delivered motivation and method design (as acknowledged by Reviewer #1). Moreover, we have now further improved this work as suggested by Reviewer #3 (see the comments 4 and 5 of Reviewer #3) by giving more thorough analysis on the effectiveness of the proposed search space and attention guided search.
> Given that we have done the best-attempt (as what standard researchers can/should do), we sincerely request that the reviewer could do a sufficiently thorough assessment of our work for making a qualified recommendation. We consider that this work brings a novel dimension and space to NAS, beyond a specific method and model performance.
>
> [1] Liu et.al. Darts: Differentiable architecture search. In ICLR, 2019.

---

### Decision · Program_Chairs · 2019-12-19

**Decision:**

Reject

**Comment:**

This paper proposes an extension of the search space of neural architecture search to include dynamic convolutions, teacher nets among others. The method is evaluated on CIFAR-10 and Imagenet with a similar setup as other architecture search methods. The reviewers found that the results did not convincingly show that the proposed improvements were better than other ways of improving neural architecture search such as rroxylessNAS.